# Functional Additives in a Selected European Sea Bass (*Dicentrarchus labrax*) Genotype: Effects on the Stress Response and Gill Antioxidant Response to Hydrogen Peroxide (H_2_O_2_) Treatment

**DOI:** 10.3390/ani13142265

**Published:** 2023-07-11

**Authors:** Antonio Serradell, Daniel Montero, Genciana Terova, Simona Rimoldi, Alex Makol, Félix Acosta, Aline Bajek, Pierrick Haffray, François Allal, Silvia Torrecillas

**Affiliations:** 1Grupo de Investigación en Acuicultura (GIA), IU-ECOAQUA, Universidad de las Palmas de Gran Canaria, Crta. Taliarte s/n, 35214 Telde, Las Palmas, Spain; 2Department of Biotechnology and Life Sciences, University of Insubria, Via J.H. Dunant, 3, 21100 Varese, Italy; genciana.terova@uninsubria.it (G.T.);; 3Global Solution Aquaculture Unit, Delacon Biotechnik Gmbh, 4209 Engerwitzdorf, Austria; 4Ecloserie Marine de Graveline Ichtus, Route des Enrochements, 59820 Gravelines, France; 5SYSAAF, French Association of Poultry and Aquaculture Breeders, Campus de Beaulieu, 35042 Rennes, France; 6MARBEC, University of Montpellier, CNRS, Ifremer, IRD, 34250 Palavas-les-Flots, France

**Keywords:** phytogenics, galactomannan-oligosaccharides, selective breeding, European sea bass (*Dicentrarchus labrax*), oxidative stress, stress response

## Abstract

**Simple Summary:**

Husbandry practices in aquaculture production may lead to stress processes and oxidative stress damages on fish tissues. Functional ingredients have profiled as suitable candidates for reinforcing the fish antioxidant response and stress tolerance. In addition, selective breeding strategies have also demonstrated a correlation between fish growth and stress reactiveness, which may be a key component in species domestication. The present study evaluates the potential of three different functional additives for gill endogenous antioxidant capacity and stress relief in a growth selected genotype of European sea bass (*Dicentrarchus labrax*) juveniles fed low-FM/FO diets. For this purpose, after 72 days of a feeding trial, all fish were subjected to an oxidative stress challenge consisting of a 1 h bath exposure to hydrogen peroxide (H_2_O_2_) at a total concentration of 50 ppm. The functional additives induced a better recovery from the stress process, with a higher reduction in fish circulating plasma cortisol 24 h after oxidative stress. In addition, the functional additives induced higher catalase gill gene expression in response to the oxidative stress insult.

**Abstract:**

Functional ingredients have profiled as suitable candidates for reinforcing the fish antioxidant response and stress tolerance. In addition, selective breeding strategies have also demonstrated a correlation between fish growth performance and susceptibility to stressful culture conditions as a key component in species domestication processes. The aim of the present study is to evaluate the ability of a selected high-growth genotype of 300 days post-hatch European sea bass (*Dicentrarchus labrax*) juveniles to use different functional additives as endogenous antioxidant capacity and stress resistance boosters when supplemented in low fish meal (FM) and fish oil (FO) diets. Three isoenergetic and isonitrogenous diets (10% FM/6% FO) were supplemented with 200 ppm of a blend of garlic and *Labiatae* plant oils (PHYTO0.02), 1000 ppm of a mixture of citrus flavonoids and *Asteraceae* and *Labiatae* plant essential oils (PHYTO0.1) or 5000 ppm of galactomannan-oligosaccharides (GMOS0.5). A reference diet was void of supplementation. The fish were fed the experimental diets for 72 days and subjected to a H_2_O_2_ exposure oxidative stress challenge. The fish stress response was evaluated through measuring the circulating plasma cortisol levels and the fish gill antioxidant response by the relative gene expression analysis of *nfΚβ2*, *il-1b*, *hif-1a*, *nd5*, *cyb*, *cox*, *sod*, *cat*, *gpx*, *tnf-1α* and *caspase 9.* After the oxidative stress challenge, the genotype origin determined the capacity of the recovery of basal cortisol levels after an acute stress response, presenting GS fish with a better pattern of recovery. All functional diets induced a significant upregulation of *cat* gill gene expression levels compared to fish fed the control diet, regardless of the fish genotype. Altogether, suggesting an increased capacity of the growth selected European sea bass genotype to cope with the potential negative side-effects associated to an H_2_O_2_ bath exposure.

## 1. Introduction

The use of biocide compounds is an extended practice in aquaculture production in order to eliminate microorganisms and other pathogenic agents in aquaculture facilities [1,2]. Among them, hydrogen peroxide (H_2_O_2_) is a powerful oxidizer compound used against fish external parasites and bacteria [3,4,5] with proven effectiveness in treating diseases in European sea bass (*Dicentrarchus labrax*) [6,7]. However, this compound is an important source of reactive oxygen species (ROS), which may induce severe tissue damages, especially on those directly exposed to the surrounding environment [8,9].

Fish gills act as a physical and biochemical semipermeable barrier with an important role in fish respiratory processes, hydromineral balance and immune responses [10,11]. In response to a stress process, such as those derived from biocides or other pollutants exposure, cortisol will target gill tissue, triggering mitochondrial rich cells (MRCs) oxidative phosphorylation (OXPHOS) in order to supply ATP to the Na^+^/K^+^ ATPase pumps involved in fish hydromineral and osmotic balance reestablishment [12,13]. During OXPHOS, some electrons may leak the electronic transport chain (ETC) prior to being reduced by the cytochrome c oxidase, reacting in the mitochondrial intermembrane with oxygen (O_2_) to form superoxide anions (O^2−^) [14,15]. Then, superoxide^-^ will be transformed by the superoxide dismutase enzyme (SOD) into H_2_O_2_, which will be finally detoxified by catalase (CAT) and glutathione peroxidase (GPX) into water and O_2_ [16]. Nevertheless, in a high-stress-susceptible fish species such as the European sea bass [17,18], the cumulative effects of both internal and external ROS increased concentrations may overwhelm fish antioxidant defense, leading to oxidative stress processes including cellular membrane lipid peroxidation and protein and DNA destruction [14,19].

Supplementing fish diets with phytogenic feed additives (PFAs) has shown potential in reinforcing the fish antioxidant status [20,21]. PFAs are plant-derived bioactive compounds with elevated contents of flavonoids, tannins and mucilages with high antioxidant properties [22,23,24]. Additionally, supplementing fish diets with PFAS has been reported to be capable of attenuating different fish species’ stress responses [25,26,27,28,29]. Plant-derived prebiotic compounds are another variety of functional additives with a potential reinforcing fish antioxidant defense [22,29]. Prebiotics have the ability to benefit the host health by selectively modulating the fish microbiome composition [30]. In previous studies, the galactomannan-oligosaccharides (GMOS) protected European sea bass juveniles’ gills against the damages derived from oxidative stress [26,31,32]. Even though a wide variety of studies report the benefits of functional additives supplementation, the mechanisms by which these compounds may favor the fish health status and welfare are still not clearly defined. Several factors can define the functional additives’ effects on fish performance, such as the different bioactive compounds’ properties, dietary inclusion levels, dietary production methodologies and the fish’s capacity to harness these products [29,31,33,34,35].

In this scenario, selective breeding has been recognized as a permanent and cumulative solution for improving fish feed efficiency and feed utilization [36,37,38], resulting in increased growth performance and better health and welfare [39]. Fish genotype selection may also contribute to increased growth performance, even coping with the nutritional variations associated with low fish meal (FM)- and fish oil (FO)-based diets [40]. Furthermore, one of the main effects of selective breeding is the fish species domestication processes by which the captive species becomes adapted to the rearing conditions [36], reducing the negative side effects associated with cultured conditions’ stress processes [41].

Accordingly, the aim of the present study was to determine the gill antioxidant capacity and stress tolerance against an H_2_O_2_ exposure oxidative stress challenge in a growth selected European sea bass genotype fed low FM/FO-based diets supplemented with three different plant-derived functional additives, PHYTO0.02, PHYTO0.01 or GMOS0.5.

## 2. Materials and Methods

### 2.1. Experimental Diets

Four low FM/FO (10%/6%) diets with isoenergetic and isonitrogenous formulations were produced by Biomar (Brande, Denmark), meeting the nutritional requirements for European sea bass juveniles [42,43]. A reference diet void of supplementation (Control), a diet supplemented with 200 ppm of a blend of garlic and *Labiatae* plant oils (87.5 mg terpenes/kg diet; PHYTO0.02), a diet supplemented with 1000 ppm of a mixture of citrus flavonoids and *Asteraceae* and *Labiatae* plant essential oils (57 mg terpenes/kg diet; PHYTO0.1) and a diet supplemented with 5000 ppm of galactomannan-oligosaccharides (GMOS0.5). The functional ingredients were supplemented according to the producer’s recommendations (Delacon, Engerwitzdorf, Austria). The PHYTO0.1 and GMOS0.05 additives were included in the mix during the pre-extrusion process in order to ensure product stability. The PHYTO0.02 additive was homogenized with the dietary oils and included by vacuum coating during the post-extrusion process (Table 1).

### 2.2. Population Design and Fish Production

The experimental design contemplated two European sea bass genotypes, a high growth selected genotype (GS) and a wild type genotype (WT), with the same scheme of selection and details previously described in [40,44].

Briefly, both genotypes were produced in the facilities of Palavas-les-flot (France) by mating 7 dams selected for growth from the MARBEC-IFREMER broodstock with 33 sires (genetically selected, GS) derived from the breeding nucleus of the EMG Ecloserie Marine de Gravelines (Gravelines, France) breeding company or 32 wild sires captured in the gulf of Lion (Wild type genotype, WT). Dams’ eggs were collected by stripping and pooled in equal representation between dams, and they were transferred into 65 tubes (one per sire). The two resulting genotypes were incubated separately at 14 °C until hatching. One-day-old hatched larvae were pooled by the equi-representation of each dam and shipped to the University of Las Palmas de Gran Canaria (ULPGC, Las Palmas de Gran Canaria, Spain) by airplane into oxygen-saturated water transport bags that were kept in insulated boxes. The larvae were grown in separated tanks following the standardized methodology of the Research Group in Aquaculture [45,46] at the ULPGC facilities. Progenies from both genotypes were kept at similar conditions during the preweaning, weaning and early juvenile growing phases.

### 2.3. Experimental Conditions

At 300 days post-hatching (dph), the fish genotype induced significant differences in fish growth. A total of 180 GS fish with a mean weight of 104.9 ± 3.1 g were randomly pooled and distributed in four 500 L tanks (30 fish/tank, 1 tank per dietary treatment). On the other hand, 360 WT fish with a mean weight of 58 ± 1.6 g were randomly pooled and distributed in twelve 500 L tanks (45 fish/tank, 3 tanks per dietary treatment). The experimental tanks presented similar initial culture densities. The tanks were supplied with filtered sea water (18.8–20 °C and 6.1–6.6 ppm dissolved oxygen) in a flow-through system under a natural photoperiod (12L:12D). The experimental diets were fed 3 times a day, 6 days a week until apparent satiation from 12 March to 29 May 2020 (72 days). The feed intake was monitored daily, and the growth performance and feed utilization were calculated at the end of the feeding experience.

At the end of the feeding experience, six fish per dietary treatment and genotype level (two fish/WT tank and six fish/GS tank) were used to obtain blood plasma samples for circulating plasma cortisol analysis and gill samples for relative gene expression analysis. This sampling point was considered as the basal point, t = 0 h (pre-stress challenge), in the statistical analysis.

### 2.4. Oxidative Stress Challenge

After 72 days of a feeding trial, experimental fish were subjected to an oxidative stress challenge consisting of a 1 h bath exposure to hydrogen peroxide (H_2_O_2_), following the procedure previously described by Roque and co-authors in 2010 [8]. Briefly, H_2_O_2_ treatment was applied by stopping experimental tanks’ water flow and aeration and adding H_2_O_2_ at a nominal concentration of 50 ppm. After 1 h of exposure, the tanks’ water flow and aeration were restored and kept at maximum renovation rate for 2 h in order to remove all the remaining H_2_O_2_.

At 2 h and 24 h after the oxidative stress challenge, six fish per dietary treatment and genotype level (two fish/WT tank and six fish/GS tank) were used to obtain blood plasma samples for circulating plasma cortisol analysis and gill samples for relative gene expression analysis.

### 2.5. Sampling Methodology

Prior to manipulation, the fish were anesthetized using diluted clove oil (diluted in ethanol 100% (1:2)) (Guinama S.L; La Pobla de Vall Bona, Valencia (46185), Spain, Ref. Mg83168) at a concentration of 0.02 mL/L.

Blood samples were obtained by a caudal sinus puncture with 1 mL syringes, stored on an heparin-coated Eppendorf and immediately centrifuged at 3000 g for 5 min at 4 °C in order to obtain plasma samples. Plasma samples were stored at −80 °C until plasmatic cortisol analysis. The plasmatic cortisol concentration was determined using the assay kit (Access Cortisol ref 33600, ©2010 Beckman Coulter, Inc.; Alcobendas, Madrid (28108), Spain) by an external laboratory Animal Lab (Las Palmas de Gran Canaria, Gran Canaria, Canary Island, Spain).

Gill samples for relative gene expression were obtained after fish euthanasia by head blow. The second and third holobranch from the fish’s left side were excised, placed in 1.5 mL Eppendorf with RNAlater and kept at 4 °C for 24 h. Afterwards, RNAlater was removed, and the samples were frozen at −80 °C until the relative gene expression analysis. RNAlater was prepared by dilution in 1 L deionized water of 650 g ammonium sulfate, 7.4 g sodium citrate dihydrate, 7.4 g EDTA di sodium salt and 200–500 µL concentrated sulfuric acid, with a final pH of 5.2, obtaining 1.4 L of RNAlater.

### 2.6. RNA Extraction and Real-Time PCR Analysis

The gill (approx. 50 mg/sample) total mRNA (ng/μL) was extracted by employing TRI-reagent (Sigma-Aldrich, Sant Louis, MO, USA) from the extraction kit RNeasy Minikit from Qiagen. An iScript^TM^ cDNA synthesis Kit (Bio-Rad, Hercules, CA, USA) was employed to perform the reverse transcriptions to obtain cDNA in a 20 μL reaction containing 1 μL of the total mRNA at a concentration of 0.5 μg/μL.

The real-time PCR analysis was performed with an iCycler with the optical module in a final volume of a 20 μL reaction, containing 10 μL iQTM-SYBER^®^ Green Supermix (Bio-Rad, Hercules, CA, USA), 5 μL of free-nuclease water, 3 μL of cDNA (1:10 dilution) and 1 μL of forward and reverse primer. The target genes were the nuclear factor kappa beta-2 (*nfκβ2*), interleukin 1β (*il1β*); hypoxia inducible factor 1α (*hif-1α*), NADH dehydrogenase subunit 5 (*nd5*), cytochrome b (*cyb*), cytochrome oxidase subunit 1 (*cox*), mitochondrial respiratory uncoupling protein 1 (*ucp1*), superoxide dismutase (*sod*), catalase (*cat*), glutathione peroxidase (*gpx*), tumor necrosis factor 1α (*tnf-1α*) and caspase 9 (*casp-9*). The specific primer sequences, annealing temperatures and accession numbers are presented in Table 2. The real-time running conditions were: 95 °C for 1 min, followed by 40 cycles at 95 °C for 10 s and an annealing temperature for 30 s (Table 2). All reactions were performed in duplicate for each sample, and a blank control containing nuclease-free water instead of cDNA in the final volume mix was included in each assay. Two constitutive genes were tested: α-tubulin *(α-tub*) and the ribosomal protein L17 (*rpl17*). Applying the CFX Maestro^TM^ Software selection tool (CFX Maestro™ Software User Guide Version 1.1, Biorad), the *α-tub* was selected as the most stable and amplification-efficient reference gene. The relative gene expression levels were calculated using the 2^−ΔΔCt^ method [47,48], using *α-tubulin* as the housekeeping gene. The gene expression was calculated relative to the transcript levels of WT fish fed the control diet at t = 0 h (pre-stress challenge).

### 2.7. Statistical Analyses

All the analyses were performed with R Project for Statistical Computing. Means and standard deviations (SD) were calculated for each parameter measured.

To assess differences in fish growth and feed utilization among the genotypes, differences in the mean specific growth rate (SGR), feed conversion ratio (FCR) and individual feed intake among the selected genotypes were tested by one-way analysis of variance (ANOVA) and a Tukey test. Similarly, to assess differences in fish growth and feed utilization among the experimental diets, differences in the mean SGR, FCR and individual feed intake among the selected experimental diets were tested by one-way analysis of variance (ANOVA) and a Tukey test. A three-way analysis of variance (ANOVA) and a Tukey test were performed to assess differences in the fish stress response and gill relative gene expression between the genotypes and experimental dietary treatments along the different sampling points.

Prior to analysis, all data were tested for outlying values through linear regression adjustment, defining the outside cut-offs as 1.5 times the Inter-Quantile Range (IQR) below the first and above the third quantiles [57,58]. Before the analysis, a Kolmogorov–Smirnov test was used to assess the quantile normality, and Levene’s test was used to assess the homogeneity of the variance. Where there was significant variance heterogeneity, the data were transformed by the square root or log transformation. When transformations did not remove the heterogeneity, the analysis was performed with untransformed data with the F-test α-value set at 0.01 [59].

## 3. Results

### 3.1. Feeding Experience

All fish grew properly along the feeding trial (72 days), presenting GS fish with significantly (*p* < 0.05) higher final body weights than those of WT fish (*p* < 0.05). GS fish presented significantly improved FCR values and lower individual feed intake values than WT fish (Table 3). Within each genotype, dietary functional additives did not affect the fish final body weight and length. No significant differences in the fish specific growth rate were found (Table 3).

### 3.2. Stress Response

At the end of the feeding experience (pre-oxidative stress challenge, t = 0 h), the GS fish presented significantly lower (*p* < 0.05) levels of mean basal circulating plasma cortisol than the WT fish. The GS fish presented a mean basal concentration of 1.7 ± 0.51 ng/mL per fish g; meanwhile, the WT fish presented a mean basal concentration of 3.67 ± 0.15 ng/ mL per fish g (Table 4).

In response to the oxidative stress challenge, at 2 h after H_2_O_2_ exposure, all the experimental groups presented a significant increase (*p* < 0.05) in circulating cortisol levels compared to the basal levels, regardless of the genotype or the dietary treatment fed. The H_2_O_2_ exposure increased the GS fish cortisol up to levels ×2.4 fold higher compared to the basal levels. Meanwhile, the WT fish presented an increase in cortisol levels of ×1.7 fold compared to their basal levels.

After 24 h of the oxidative stress challenge, all experimental groups presented a significant reduction (*p* < 0.05) in plasmatic cortisol levels down to the basal levels observed at t = 0 h pre-oxidative stress challenge, regardless of the genotype or the dietary treatment fed. The GS fish presented a decrease in cortisol levels of ×2.2 fold compared to those levels observed at 2 h post-oxidative stress challenge, whereas the WT fish cortisol levels presented a decrease of ×1.7 fold compared to those levels observed at 2 h after the oxidative stress challenge.

### 3.3. Gill Relative Gene Expression

Prior to the oxidative stress challenge (t = 0 h), the fish gill antioxidant defense-related gene expression presented significant differences (*p* < 0.05) associated with the interaction between the genotype and the dietary treatment fed (Figure 1). The GS fish fed the control and PHYTO0.1 diets showed a higher (*p* < 0.05) cat basal gill expression compared to the WT fish fed the same dietary treatments. The GS fish fed the control diet also presented upregulated (*p* < 0.05) *sod* gene expression levels compared to the WT fish fed the same diet. On the contrary, the WT fish fed the GMOS0.5 diet presented a higher (*p* < 0.05) *sod* basal gene expression than the GS fish fed the same dietary treatment.

Within the WT fish genotype, the diet fed directly affected the fish gill basal antioxidant gene expression. The fish fed the GMOS0.5 diet presented the highest (*p* < 0.05) *sod* expression levels, followed by PHYTO0.02 and PHYTO0.1, respectively. Similarly, those fish fed with PHYTO0.1 and GMOS0.5 diets presented the highest (*p* < 0.05) gpx gill expression levels. Those fish fed the GMOS0.5 diet presented significantly higher (*p* < 0.05) *hif-1α* relative expression levels than those fish fed the control and PHYTO0.1 diets (Appendix A Table A1).

Two hours post-oxidative stress challenge, a generalized upregulation of antioxidant defense-related gene expression was observed. All fish presented a significant increase (*p* < 0.05) in *cat* and *gpx* gill expression levels (Appendix A Table A1), whereas the gill *sod* gene expression presented significant differences associated with the fish genotype and the dietary treatment fed. The GS fish fed the control diet presented an upregulation (*p* < 0.05) of gill *sod* relative expression levels compared to the WT fish fed the same dietary treatment. In fact, within the GS genotype, the fish fed the control diet induced the highest (*p* < 0.05) *sod* expression levels. Accordingly, as a response to the H_2_O_2_ exposure, an up-regulation of the mitochondrial ETC-related gene expression was observed. All the experimental fish groups presented a general increase (*p* < 0.05) in *cox* transcript levels, independently of the fish origin or diet fed. However, this response was more acute for GS fish fed the PHYTO0.1 diet, which presented a higher (*p* < 0.05) expression than the WT fish fed the same diet. The GS fish fed the control diet were the only experimental group presenting an increase (*p* < 0.05) in gill *ucp1* relative gene expression in relation to their basal levels (Figure 2).

In regard to the results observed on genes related with a proinflammatory response, 2 h after the oxidative stress challenge, all the experimental groups presented a significantly increased (*p* < 0.05) *nfΚβ2* gill gene expression. Only the WT fish fed the PHYTO0.2 and GMOS0.5 diets presented significantly increased (*p* < 0.05) il-1*β* gill transcription levels in relation to basal levels. At this sampling point, feeding a GMOS0.5 diet to WT fish resulted in an increase (*p* < 0.05) in the *casp-9* gill relative gene expression, whereas the GS fish fed the GMOS0.5 diet presented an increased (*p* < 0.05) *hif-1α* gill relative gene expression.

Twenty-four hours after the oxidative stress challenge, the GS fish fed the control diet and the WT fish fed the GMOS0.5 diet presented a downregulation (*p* < 0.05) of *sod* and *gpx* gill relative gene expression, respectively. Despite the changes mentioned above, no significant differences were found among the different experimental groups in terms of the antioxidant defense gene response (Figure 3).

At this sampling point, the GS fish fed the PHYTO0.1 diet presented a significant downregulation (*p* < 0.05) of gill *cox* relative gene expression. No differences among the groups were observed in the fish cox gill gene expression. On the contrary, the WT fish fed the PHYTO0.1 diet presented higher (*p* < 0.05) *ucp1* gill relative gene expressions than the WT fish fed the GMOS0.5 diet.

At the end of the oxidative stress challenge, no significant change in the fish gill pro-inflammatory gene response was observed, with an exception for the il-1*β* gill relative gene expression, which was increased (*p* < 0.05) in the WT fish fed the control diet in relation to the previous sampling point (Appendix A Table A1).

## 4. Discussion

The results of the present study highlight the strong effect exerted by breeding selection, leading to a two-fold higher body weight for GS fish compared to WT at the same age, 300 days post-hatching. At the end of the feeding trial, both genotypes presented proper growth, almost doubling the initial body weight regardless of the dietary treatment fed. Despite the fish from both genotypes presenting similar specific growth rates, the GS fish presented improved feed conversion ratios and lower individual feed intakes than the WT fish, indicating a better capacity to harness feed even when dealing with low FM/FO-based diets. In agreement with these results, in the study carried out by Montero and co-authors in 2023 [40], GS fish belonging to the same breeding program presented a higher growth performance and a better plasticity to cope with the possible nutritional imbalances derived from low FM/FO-based diets. At the end of the feeding experience, GS fish presented a higher body weight, decreased fish perivisceral fat deposition and increased flesh DHA and ARA contents compared to WT fish.

The use of selective breeding strategies as a tool to increase fish growth performance may lead to favoring the selection of secondary functional phenotypes, such as stress tolerance and behavioral traits [38], which are keystones in domestication processes [60]. In 2016, Vandeputte and co-authors [61] studied the stress response of three different genotypes of European sea bass (wild, domesticated and selected for growth) subjected to acute confinement followed by a swimming stress challenge. The authors reported a negative correlation between the fish body weight and the circulating plasma cortisol levels after the stress challenge, concluding that selective breeding may favor fish’s low stress responsiveness. Accordingly, in the present study, the GS fish presented significantly lower basal cortisol levels than the WT fish, pointing to a possible effect of growth selective breeding on fish stress indicators. Furthermore, and despite presenting higher cortisol levels than the WT fish in the first hours after the oxidative stress challenge, the GS fish presented a better recovery back to the basal cortisol levels at 24 h post-H_2_O_2_ exposure. A better competence for recovering a homeostatic status might be advantageous under aquaculture conditions in which fish are constantly exposed to stressful conditions [17,18]. An effective and controlled physiological stress response will avoid the negative side-effects associated with a chronic cortisol exposure [39,62]. An example of stress tolerance benefits for aquaculture production was reported by Øverli and co-authors in 2006 [63]. The authors studied the effects of a transport stress challenge on the feed utilization of two different genotypes of rainbow trout (*Oncorhynchus mykiss*) selected for low or high stress responsiveness. The low stress responsive genotype presented a significantly higher feed efficiency and a lower food waste production after the stress challenge.

In the present study, the genetic selection also induced differences in fish antioxidant defense gene expression. At the basal level, at t = 0 h pre-stress challenge, the GS fish presented higher *cat* gene expression levels than the WT fish. Similarly, other studies have reported higher antioxidant defenses in the selected genotypes of other fish species. For example, Solberg and co-authors, in 2012 [64], described higher glutathione reductase, Cu/Zn *sod* and *gpx* relative gene expression levels in response to environmental stress processes for a domesticated strain of Atlantic salmon (*Salmo salar*) in comparison to a wild strain. In 2010, Sauvage and co-authors [65] observed a higher gene expression of three genes associated with protective properties against oxidative stress processes (precursor of hemopexin, heme-binding protein 2, precursor of fibrinogen γ chain and precursor of the inter-α trypsin inhibitor heavy chain H2) on a selected strain of Brook charr (*Salvelinus fontinalis*) (F4 generation) compared to a reference population obtained from randomly mixed breeders (F1 generation) kept at the same environmental conditions. Palinska-Zarska and co-authors, in 2021 [66], compared the antioxidant enzymatic activity of two genotypes, domesticated and wild, of perch (*Perca fluviatilis*) larvae presenting higher *sod* and *cat* activities than the domesticated strain. These authors suggested that a higher antioxidant enzyme activity in the selected strain resulted in a better adaptation to the formulated feed, leading to better survival rates and performance during the larval weaning period. In the present study, the functional additives also presented an effect on fish antioxidant defense. The WT fish fed functional diets presented higher *sod*, *cat* and *gpx* basal gene expression levels than the WT fish fed the control diet. In the same way, at two hours after H_2_O_2_ exposure, those fish fed the functional additives presented the highest *cat* gene expression levels compared to the fish fed the control diet, regardless of the genotype. This could suggest an enhanced antioxidant capacity associated with functional additives supplementation, as catalase is the main enzyme contributing to H_2_O_2_ removal when found in high concentrations in the intercellular space [66]. Dietary supplementation with plant origin compounds may reinforce the fish antioxidant status through the interaction with several signaling transcription factors modulating fish antioxidant-related gene expression [33]. Li and co-authors, in 2018 [67], evaluated the effects of pinostrobin, a potent flavonoid extracted from pines, on zebra fish’s (*Danio rerio*) neural antioxidant status. This phytogenic compound increased fish GSH-PX, GSH/GSSG, SOD and CAT enzymes, reducing fish neural oxidative stress damages and apoptotic processes. Mansour and co-authors, in 2020 [68], analyzed the antioxidant capacity of sea bream (*Sparus aurata*) fed diets supplemented with Moringa (*Moringa oleifera*) against an H_2_O_2_ exposure at a concentration of 50 ppm. The authors reported an enhanced response of the fish fed the supplemented diets, with an increased gill *cat* gene expression compared to that of those fish fed diets void of supplementation. In addition, these compounds are rich in terpenes and flavonoids, which present high antioxidant properties preventing the formation or directly quenching the oxygen and nitrogen reactive species derived from aerobic metabolism [22,69].

An increased aerobic metabolism rate, in order to respond against a stress process, may also suppose an important source of oxidative stress processes. During oxidative phosphorylation, between 1 and 3% of all electrons may “leak” from the electron transport chain [14] being released into the mitochondrial intermembrane space, where they will react with O_2_ generating ROS. In the present study, the stress challenge resulted in a generalized overexpression of the ETC-related genes *nd5*, *cyb* and *cox*, regardless of the fish genotype or the dietary treatment fed. However, an interesting response was observed for GS fish fed the control diet, which, unless presenting similar levels of expression as the other experimental groups, presented an increased *ucp1* gene expression after H_2_O_2_ exposure. In the absence of an external surplus of antioxidant defenses such as the antioxidant properties of functional additives, this may suggest a feedback mechanism limiting mitochondrial ROS formation, in a process called “uncoupling to survive” [15], and protecting gill tissue from oxidative stress processes.

ROS are important metabolic agents involved in fish inflammatory responses through the interaction with the nuclear factor kappa beta (NFKβ) [70,71] and leading to the activation of the pro-inflammatory cytokines IL-1β and TNF-α [72]. In the present study, all experimental treatments presented similar expressions of pro-inflammatory genes in gills after H_2_O_2_ exposure. Nevertheless, the GS fish fed the GMOS0.5 diet presented upregulated *hif-1α* gill expression levels compared to the fish fed the rest of the dietary treatments. Under hypoxic conditions associated with inflammatory processes [73], the *hif-1α* mediates the activation of the O_2_-independent glycolytic pathway, ensuring ATP production to cope with the bio-energetic requirements [74,75]. On the other hand, the WT fish fed GMOS0.5, which did not present an increased expression of *hif-1α*, presented an increased expression of *caspase 9*, which is the activator of caspase-dependent apoptotic processes [76], suggesting a lower ability to cope with the side-effects associated with the inflammatory process.

## 5. Conclusions

In conclusion, H_2_O_2_ exposure induced the triggering of both the fish stress response and oxidative stress defense. The GS genotype fish presented a better capacity to recover the basal cortisol levels, suggesting a higher tolerance to potential stressful scenarios associated with fish rearing conditions. In addition, the use of functional additives enhanced the fish antioxidant response via upregulating the expression of *cat* in gill expression levels in response to the oxidative insult. The GMOS0.5 diet induced the activation of hif-1α gene expression in the gills of GS fish, modulating the triggering of pro-inflammatory-associated processes. Nevertheless, in the view of the complexity of interactions between fish genetic traits and the diversity of functional ingredients, more experiences must be carried out to address the best nutritional and genetic selection strategies in order to promote fish health and welfare under rearing conditions.

## Figures and Tables

**Figure 1 animals-13-02265-f001:**
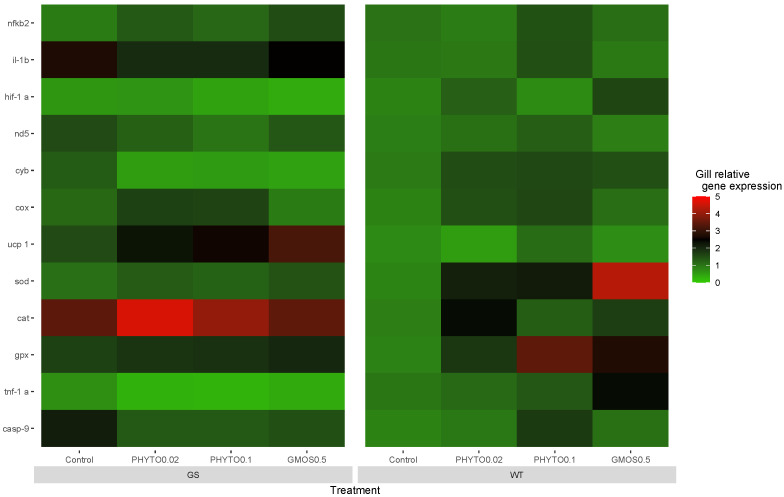
European sea bass gill relative gene expression heat map at 0 h pre-oxidative stress challenge for high-growth selected genotype (GS) and wild type genotype (WT) European sea bass. Control (Control diet); PHYTO0.02 (PHYTO0.02 diet, supplemented with a 200 ppm blend of phytogenic feed additives consisting of a mixture of garlic and *Labiatae* plant essential oils with 87.5 mg terpens/kg diet); PHYTO0.1 (PHYTO0.1 diet, supplemented with a 1000 ppm blend of phytogenic feed additives, consisting of a mixture of citrus fruits and *Asteraceae* and *Labiatae* plant essential oils with 57 mg terpens/kg diet); GMOS0.5 (GMOS0.5 diet; supplemented with 5000 ppm galactomannan-oligosaccharides).

**Figure 2 animals-13-02265-f002:**
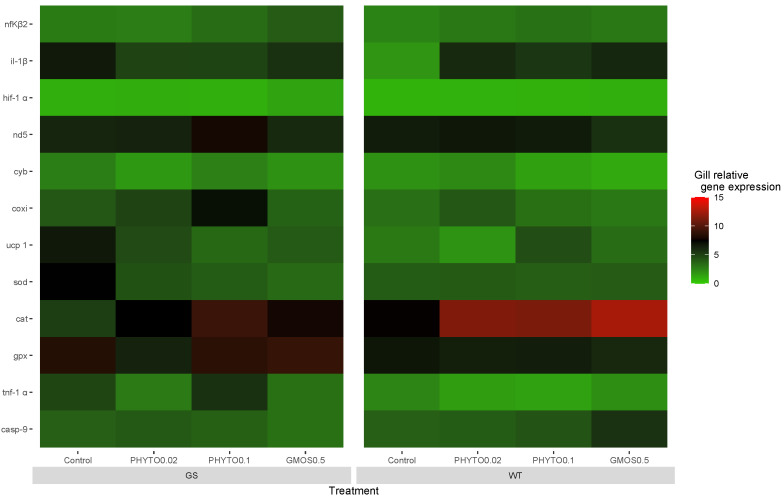
European sea bass gill relative gene expression heat map at 2 h after oxidative stress challenge for high-growth selected genotype (GS) and wild type genotype (WT) European sea bass. Control (Control diet); PHYTO0.02 (PHYTO0.02 diet, supplemented with a 200 ppm blend of phytogenic feed additives consisting of a mixture of garlic and *Labiatae* plant essential oils with 87.5 mg terpens/kg diet); PHYTO0.1 (PHYTO0.1 diet, supplemented with a 1000 ppm blend of phytogenic feed additives, consisting of a mixture of citrus fruits and *Asteraceae* and *Labiatae* plant essential oils with 57 mg terpens/kg diet); GMOS0.5 (GMOS0.5 diet; supplemented with 5000 ppm galactomannan-oligosaccharides).

**Figure 3 animals-13-02265-f003:**
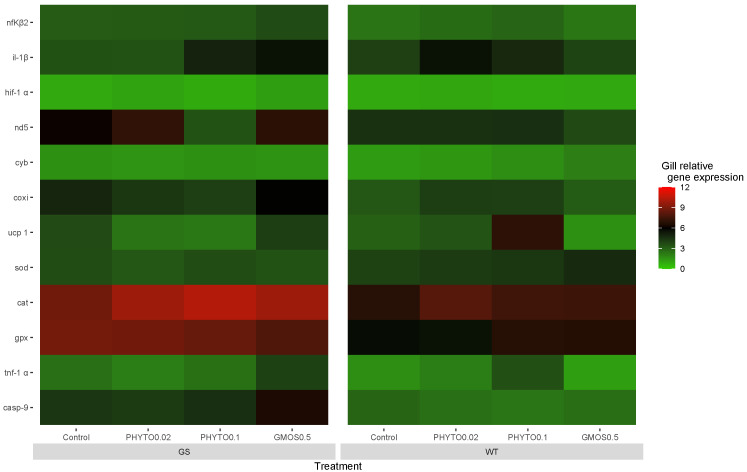
European sea bass gill relative gene expression heat map at 24 h after oxidative stress challenge for high-growth selected genotype (GS) and wild type genotype (WT) European sea bass. Control (Control diet); PHYTO0.02 (PHYTO0.02 diet, supplemented with a 200 ppm blend of phytogenic feed additives consisting of a mixture of garlic and *Labiatae* plant essential oils with 87.5 mg terpens/kg diet); PHYTO0.1 (PHYTO0.1 diet, supplemented with a 1000 ppm blend of phytogenic feed additives, consisting of a mixture of citrus fruits and *Asteraceae* and *Labiatae* plant essential oils with 57 mg terpens/kg diet); GMOS0.5 (GMOS0.5 diet; supplemented with 5000 ppm galactomannan-oligosaccharides).

**Table 1 animals-13-02265-t001:** Main ingredients and analyzed proximal composition of the experimental diets.

Ingredients	Diet (%)			
	Control	PHYTO0.02	PHYTO0.1	GMOS0.5
Fish meal ^1^	9.6	9.6	9.6	9.6
Soya protein concentrate	18.2	18.2	18.2	18.2
Soya meal	11.6	11.6	11.6	11.6
Corn gluten meal	24.1	24.1	24.1	24.1
Wheat	8.585	8.565	8.485	8.085
Wheat gluten	1.9	1.9	1.9	1.9
Guar meal	7.7	7.7	7.7	7.7
Rapeseed extracted	3.0	3.0	3.0	3.0
Fish oil ^2^	6.5	6.5	6.5	6.5
Rapeseed oil ^3^	5.2	5.2	5.2	5.2
Vitamin and mineral premix ^4^	3.6	3.6	3.6	3.6
Antioxidant ^5^	0.015	0.015	0.015	0.015
Phytogenic (garlic and *Labiatae* plant essential oils) ^6^	0	0.02	0	0
Phytogenic (citrus fruits and *Asteraceae* and *Labiatae* plant essential oils) ^7^	0	0	0.1	0
Galactomannan-oligosaccharides (GMOS) ^8^	0	0	0	0.5
Proximate composition (% of dry matter)				
Crude lipids	19.91	20.44	20.44	20.47
Crude protein	49.30	49.27	49.27	49.76
Moisture	5.10	5.01	5.01	5.06
Ash	7.02	6.41	6.41	6.49

Dietary ingredient composition and proximal composition expressed as % of dry weight. Control (Control diet), PHYTO0.02 (PHYTO diet, 200 ppm mixture of garlic and *Labiatae* plant essential oils), PHYTO0.1 (PHYTO diet, 1000 ppm mixture of citrus fruits and *Asteraceae* and *Labiatae* plant essential oils), GMOS (GMOS diet, 5000 ppm galactomannan-oligosaccharides). ^1^ South American, Superprime 68%. ^2^ South American fish oil. ^3^ DLG AS, Denmark. ^4^ Vilomix, Denmark. ^5^ BAROX BECP, BHT. ^6^ Delacon Biotechnik GmbH, Austria. ^7^ Delacon Biotechnik GmbH, Austria. ^8^ Delacon Biotechnik GmbH, Austria.

**Table 2 animals-13-02265-t002:** Primer sequences of the different genes analyzed and their RT-PCR conditions.

Gene	Access. Number	Primer	Nucleotide Sequence 5′-3′	Annealing T (°C)	Reference
*nfΚβ2*	KM225790	Fw	CTGGAGGAAACTGGCGGAGAAGC	60	[49]
		Rv	CAGGTACAGGTGAGTCAGCGTCATC		
*il-1b*	AJ53742	Fw	ATTACCCACCACCCACTGAC	60	[50]
		Rv	TCTCTTCCACTATGCTCTCCAG		
*hif-1a*	DQ171936	Fw	GACTTCAGCTGCCCTGATTC	60	[51]
		Rv	GGCTGGTTTATAGCGCTGAG		
*nd5*	KF857307	Fw	CCCGATTTCTGTGCCCTACTA	60	[52]
		Rv	AGGAAAGGAGTGCCTGTGA		
*cyb*	EF427553	Fw	TGCCTACGCTTCCTTCGCTCGATCC	60	[53]
		Rv	TAACGCCAACACCCCGCCCAAT		
*cox*	KF857308	Fw	ATACTTCACATCCGCAACCATAA	60	[53]
		Rv	AAGCCTCCGACTGTAAATAAGAAA		
*ucp1*	MH138003	Fw	CGATTCCAAGCCCAGACGAACCT	60	[53]
		Rv	TGCCAGTGTAGCGACGAGCC		
*sod*	FJ860004.1	Fw	CATGTTGGAGACCTGGGAGA	60	[54]
		Rv	TGAGCATCTTGTCCGTGATGT		
*cat*	FJ860003.1	Fw	TGGGACTTCTGGAGCCTGAG	60	[54]
		Rv	GCAAACCTCGATCGCTGAAC		
*gpx*	FM013606.1	Fw	AGTTCGTGCAGTTAATCCGGA	60	[54]
		Rv	GCTTAGCTGTCAGGTCGTAAAAC		
*tnf-1α*	DQ200910.1	Fw	GCCAAGCAAACAGCAGGAC	60	[52]
		Rv	ACAGCGGATATGGACGGTG		
*casp-9*	DQ345775	Fw	GGCAGGACTCGACGAGATAG	62.7	[55]
		Rv	CTCGCTCTGAGGAGCAAACT		
*α-tub (hk)*	AY326429.1	Fw	AGGCTCATTGGCCAGATTGT	60	[31]
		Rv	CAACATTCAGGGCTCCATCA		
*rpl17*	AF139590	Fw	GAGGACGTGGTGGTTCATCT	60	[56]
		Rv	CTGGCTTGCCTTTCTTGACT		

Fw: Forward primer sequence, Rv: Reverse primer sequence.

**Table 3 animals-13-02265-t003:** Growth parameters and feed utilization of European sea bass (*Dicentrarchus labrax*) juveniles (at age 372 dph) after 72 days of the feeding experience.

	WT Genotype				GS Genotype					
Diet	Control	PHYTO0.02	PHYTO0.1	GMOS0.5	Control	PHYTO0.02	PHYTO0.1	GMOS0.5		
									One-way ANOVA	
									Diet (Inside Each Genotype)	
IBW (g) (300 dph)	58 ± 9.2	57.8 ± 10.2	58.6 ± 10	57.5 ± 9.4	108.7 ± 15.4	106.2 ± 17.1	102.2 ± 17	102.4 ± 15.6	ns	
IL (300 dph)	17.6 ± 1	17.6 ± 0.9	17.7 ± 1	17.6 ± 0.87	20.8 ± 1.1	20.9 ± 1.1	20.5 ± 1.2	20.6 ± 1.3	ns	
FBW (g) (372 dps)	99.4 ± 18.3	95.3 ± 18.2	103 ± 19	99.8 ± 18.1	192.8 ± 31.7	189.7 ± 34	176.1 ± 33.2	180.4 ± 30.3	ns	
FL (cm) (372 dph)	21 ± 2.1	20.8 ± 1.2	21 ± 2.1	21.1 ± 1.2	25.6 ± 1.1	25.5 ± 1.3	25.1 ± 1.3	24.8 ± 1.4	ns	
									One-way ANOVA	
									Diet	Genotype
^1^ SGR (%/day)	0.75 ± 0.03	0.70 ± 0.06	0.78 ± 0.04	0.76 ± 0.01	0.80 ± 0.01	0.83 ± 0.01	0.81 ± 0.01	0.80 ± 0.01	ns	ns
^2^ FCR	1.84 ^a^ ± 0.16	1.99 ^a^ ± 0.29	1.78 ^a^ ± 0.17	1.70 ^a^ ± 0.12	1.48 ^b^ ± 0.01	1.46 ^b^ ± 0.01	1.58 ^b^ ± 0.01	1.58 ^b^ ± 0.01	ns	F = 8.335, *p*-val = 0.0119
^3^ FI (g feed/ 100 g BW/day)	0.48 ^a^ ± 0.02	0.48 ^a^ ± 0.02	0.47 ^a^ ± 0.02	0.46 ^a^ ± 0.01	0.27 ^b^ ± 0.00	0.28 ^b^ ± 0.00	0.27 ^b^ ± 0.00	0.29 ^b^ ± 0.00	ns	F = 364.1, *p*-val = 2.03 × 10^−11^

Different lowercase letters denote significant differences (*p* < 0.05) between genotypes in each sampling point (three-way ANOVA: Genotype × Diet × Time; Tukey post hoc test). ns = not significant. Values expressed in the mean ± SD. Control (Control diet); PHYTO0.02 (PHYTO0.02 diet, supplemented with a 200 ppm blend of phytogenic feed additives consisting of a mixture of garlic and *Labiatae* plant essential oils with 87.5 mg terpens/kg diet); PHYTO0.1 (PHYTO0.1 diet, supplemented with a 1000 ppm blend of phytogenic feed additives, consisting of a mixture of citrus fruits and *Asteraceae* and *Labiatae* plant essential oils with 57 mg terpens/kg diet); GMOS0.5 (GMOS0.5 diet; supplemented with 5000 ppm galactomannan-oligosaccharides); GS (high-growth selected genotype); WT (wild type genotype); IBW (initial body weight (g)); FBW (final body weight (g) 72 days after feeding experience); FL (final length (g) 72 days after feeding experience); SGR (specific growth rate 72 days after feeding experience). ^1^ SGR = [(ln average final body weight − ln average initial body weight/no days] × 100. ^2^ FCR = Feed consumption (g)/weight gain (g). ^3^ FI = Individual feed intake (g).

**Table 4 animals-13-02265-t004:** Circulating plasma cortisol level expressed in ng/mL per fish g of European sea bass (*Dicentrarchus labrax*) juveniles at t = 0 h pre-oxidative stress challenge and at t = 2 h and 24 h after the oxidative stress challenge.

	WT Genotype				GS Genotype			
	Control	PHYTO0.02	PHYTO0.1	GMOS0.5	Control	PHYTO0.02	PHYTO0.1	GMOS0.5
Time								
0 h	3.49 ^a1^ ± 1.10	3.77 ^a1^ ± 0.62	3.61 ^a1^ ± 0.92	3.83 ^a1^ ± 1.15	0.92 ^b1^ ± 0.23	1.96 ^b1^ ± 0.89	1.95 ^b1^ ± 0.21	1.97 ^b1^ ± 0.85
2 h	7.22 ^a2^ ± 2.95	5.80 ^a2^ ± 1.62	5.51 ^a2^ ± 1.29	6.20 ^a2^ ± 1.13	3.43 ^b2^ ± 0.53	3.43 ^b2^ ± 0.53	4.23 ^b2^ ± 1.43	3.81 ^b2^ ± 0.24
24 h	3.26 ^a1^ ± 0.31	3.54 ^a1^ ± 1.35	3.57 ^a1^ ± 1.73	4.10 ^a1^ ± 1.32	2.14 ^b1^ ± 0.41	1.61 ^b1^ ± 0.47	1.58 ^b1^ ± 0.13	1.79 ^b1^ ± 0.68
	Three-way ANOVA							
		Diet	Genotype	Time	D × G	D × T	G × T	D × G × T
Plasmatic cortisol		ns	F = 0.41; *p*-val = 2 × 10^−16^	F = 55.023; *p*-val = 3.16 × 10^−16^	ns	ns	ns	ns

Different lowercase letters denote significant differences (*p* < 0.05) between genotypes in each sampling point (three-way ANOVA: Diet × Genotype × Time; Tukey post hoc test). Different numbers denote significant differences (*p* < 0.05) between experimental sampling points (three-way ANOVA: Genotype × Diet × Time; Tukey post hoc test). ns = not significant. Values expressed in mean ± SD. Control (Control diet); PHYTO0.02 (PHYTO0.02 diet, supplemented with a 200 ppm blend of phytogenic feed additives consisting of a mixture of garlic and *Labiatae* plant essential oils with 87.5 mg terpens/kg diet); PHYTO0.1 (PHYTO0.1 diet, supplemented with a 1000 ppm blend of phytogenic feed additives, consisting of a mixture of citrus fruits and *Asteraceae* and *Labiatae* plant essential oils with 57 mg terpens/kg diet); GMOS0.5 (GMOS0.5 diet; supplemented with 5000 ppm galactomannan-oligosaccharides); GS (high-growth selected genotype); WT (wild type genotype).

## Data Availability

Not applicable.

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
