# Peer review of "Functional Additives in a Selected European Sea Bass (Dicentrarchus labrax) Genotype: Effects on the Stress Response and Gill Antioxidant Response to Hydrogen Peroxide (H2O2) Treatment"

_animals, 2023, doi:10.3390/ani13142265_

Round 1

Reviewer 1 Report

The present study examines an interesting topic regarding the effects of feed functional plant additives on the growth, and stress response of selected European sea bass (Dicentrarchus labrax) genotype after hydrogen peroxide treatment.

These studies are appropriate for the Animals Journal. The paper is quite well-written and has its merits.

However, I have major concerns that must be considered and addressed by the authors before I can recommend the paper for publication. My major doubts and concerns are about the correct statistical analysis of the experimental data. All the comparisons among the biological indicators (plasma cortisol, fish gill gene expression analyses) to assess fish stress response between the two populations of wild type (WT) and growth selected genotype (GS) were conducted in fish of the same age but with different initial body weight, almost double for GS compared to WT (58 g for WT, 105 g for GS). Although the authors stated that fish weight was used as a co-variable for all data analyzed (lines 166-167), which means that they have realized that is expected to affect the outcome of the dependent variables in their study, analysis of covariance did not apply to their data according to the description of Statistical analyses in section 2.6 or as presented in the Results Tables. Even within the GS genotype, the initial fish weights among the different feeding treatments are markedly not alike (e.g., 108.7 in the Control vs 102.2 and 102.4 in PHYTO0.1 AND GMOS0.5, respectively). The authors performed a one-way ANOVA analysis on fish growth performance indices and a three-way ANOVA to analyze fish stress response and gill gene expression having diet, genotype, and time as fixed factors.

Authors are advised to use a professional statistician to apply appropriate statistical methods to produce trustworthy results, reformulate the Tables and discuss whether the growth covariate affected the results. This would be a key compiled information.

Another point for criticism is the different initial fish densities (30 fish/tank for WT vs 45 fish/tank in GS) as well as the different number of tanks for each genotype used for contacting the trial (12 tanks for WT vs 4 thanks for GS).

Why authors did not employ other stress blood indicators, for example, lactate and glucose?

The above points should be clarified in the revised manuscript.

Other comments

Line 71: 'an intense oxidative agent'

Line 81: 'has been reported'

Line 97: Reference# 34 refers to dairy cattle, not fish. Please replace this reference with a reference relevant to fish.

Lines 104-108: Improve English meaning

Line 122: Please mention the temperature range during the extrusion process for PHYTO0.1 and GMOS0.05 diets

Line 136: The EU has suspended authorization for the use of ethoxyquin as a feed additive, so please explain why you used this additive.

Line 150: Please clarify how 6-7 generations correspond to more than 35 years of selection efforts to facilitate the reader who is not familiar with the breeding selection

Lines 173-175: Please refer to how many fish per tank were sampled.

Line 181: 'and aeration'

Minor editing of English language required.

Author Response

     We kindly thank the reviewer commentaries of the preliminary version of the present study, an important effort is going to be carried out in order to improve the expression of the different section of manuscript.

  1. All the comparisons among the biological indicators (plasma cortisol, fish gill gene expression analyses) to assess fish stress response between the two populations of wild type (WT) and growth selected genotype (GS) were conducted in fish of the same age but with different initial body weight, almost double for GS compared to WT (58 g for WT, 105 g for GS). Although the authors stated that fish weight was used as a co-variable for all data analyzed (lines 166-167), which means that they have realized that is expected to affect the outcome of the dependent variables in their study, analysis of covariance did not apply to their data according to the description of Statistical analyses in section 2.6 or as presented in the Results Tables. Even within the GS genotype, the initial fish weights among the different feeding treatments are markedly not alike (e.g., 108.7 in the Control vs 102.2 and 102.4 in PHYTO0.1 AND GMOS0.5, respectively). The authors performed a one-way ANOVA analysis on fish growth performance indices and a three-way ANOVA to analyze fish stress response and gill gene expression having diet, genotype, and time as fixed factors.

We must clarify that the main objective of the present study was not to evaluate the genetic selection effects on fish growth and feed utilization, but to evaluate genetic selection effects on fish stress tolerance and ability to harness different functional additives with anti-stress and anti-oxidant properties. Montero and co-authors previously reported the effects of this particular breeding program on fish growth and feed utilization [Montero et al., 2023].

  • Montero, D., Carvalho, M., Terova, G., Fontanillas, R., Serradell, A., Ginés, R., Tuse, V., Acosta, F., Rimoldi, S., Bajek, A., Haffray, P., Allal, F. & Torrecillas, S. (2023). Nutritional innovations in superior European sea bass (Dicentrarchus labrax) genotypes: Implications on fish performance and feed utilization. Aquaculture572, 739486.

     In the present study, all the experimental fish presented the same age pointing per se the effectiveness of fish genetic selection on their growth performance. The experience was designed in order to establish the same density culture conditions for both genotypes.

  • WT genotype – 360 fish distributed in 12 tanks (n= 3 tanks/ dietary treatment; 45 fish per tank)
  • GS genotype – 180 fish distributed in 4 tanks (n= 1 tank/ dietary treatment; 30 fish per tank)

     In the beginning of the feeding trial, inside each genotype level (GS or WT) no significant differences were found on fish initial body weight between the different experimental tanks.

  • WT IBW – F= 1.045 p-val= 0.405
  • GS IBW – F= 1.121 p-val= 0.344

Thus, the possible effects of the different dietary treatments on fish growth and feed utilization could be isolated in posterior analysis

     At the end of the feeding experience, inside each genotype, no significant effects of dietary treatment were found on fish final body weight (Image 2).

  • WT FBW – F= 2.306 p-val= 0.0758
  • GS FBW – F= 1.133 p-val= 0.339

     Regarding the possibility of performing Two-way ANCOVA or a TWO-way ANOVA analysis in order to evaluate fish performance, the statistical category GS only presented one tank per dietary treatment (n=1) making impossible the utilization of these statistical analysis. In addition, fish were not individually tagged at the beginning of the feeding trial (as it would suppose an elevated cost to analyse a secondary objective in the present study) making impossible to track fish individual performance.

     Thus, fish growth performance and feed utilization was analyzed by SGR, FCR and feed intake (total FI and Individual FI calculated as: total feed consumption/ tank final fish number) individually for each statistical category (genotype or dietary treatment). The statistical analysis did not detected significant differences on fish SGR neither associated to fish genotype or the dietary treatment fed. Meanwhile, both FCR and FI presented significant differences associated to fish genotype. This results will be included and discussed in the revised version of the manuscript.

 SGR

Df

Sum Sq

Mean Sq

F value

Pr(>F)

genotype

1

0.0063

0.006302

1.662

0.218

Residuals

14

0.05309

0.003792

 SGR

Df

Sum Sq

Mean Sq

F value

Pr(>F)

diet

3

0.00632

0.002106

0.192

0.9

Residuals

12

0.13142

0.010952

 FCR

Df

Sum Sq

Mean Sq

F value

Pr(>F)

genotype

1

0.2626

0.2626

8.335

0.0119

Residuals

14

0.441

0.0315

GS gnotpe

WT  genotype

FCR mean

=

1.53

1.82

FCR sd

=

0.06

0.2

 FCR

Df

Sum Sq

Mean Sq

F value

Pr(>F)

diet

3

0.0724

0.02412

0.459

0.716

Residuals

12

0.6312

0.0526

 Total FI

Df

Sum Sq

Mean Sq

F value

Pr(>F)

genotype

1

315006

315006

17.9

0.000839

Residuals

14

246369

17598

GS genotype

WT genotype

Total FI mean

=

3736.235

3412.195

Total FI sd

=

103.1824

139.6194

 Total FI

Df

Sum Sq

Mean Sq

F value

Pr(>F)

diet

13427

4476

0.098

0.96

Residuals

547948

45662

 Indiv FI

Df

Sum Sq

Mean Sq

F value

Pr(>F)

genotype

1

7499

7499

1105

1.01E-14

Residuals

14

95

7

GS genotype

WT genotype

Indiv. FI mean

=

126.68

76.68

Indiv. FI sd

=

 1.58

2.82

 Indiv FI

Df

Sum Sq

Mean Sq

F value

Pr(>F)

diet

3

8

2.7

0.004

0.99

Residuals

12

7586

632.2

     Regarding fish stress response, circulating plasma cortisol levels presented a higher statistical robustness with an n=6 genotype/diet (WT fish - 2 samples per tank - / GS fish - 6 samples per tank -) allowing the correct performance of a Three-way ANOVA with the orthogonal fixed factors GENOTYPE x DIET x TIME. In the revised version of the manuscript, the cortisol will be expressed in ng/ml of plasmatic cortisol per fish gram (g) eliminating the effects of fish size on the stress response analysis.

     Finally, the gill relative gene expression analysis were statistically robust and did not need normalization in function of fish weight as they were relative gene expression analysis ad not total gene expression analysis [Schmittgen and Livak, 2008]. The results were calculated employing Livak and Schmittgen 2-ΔΔCt method [Livak and Schittgen, 2001], in relation to the gill gene expression obtained for WT fish fed the Control diet. The three-way ANOVA with the orthogonal fixed factors GENOTYPE x DIET x TIME was robust and correctly applied.

  • Schmittgen, T. D., & Livak, K. J. (2008). Analyzing real-time PCR data by the comparative CT method. Nature protocols, 3(6), 1101-1108. https://doi.org/10.1038/nprot.2008.73

  • Livak, K.J.; Schmittgen, T.D. Analysis of relative gene expression data using real-time quantitative PCR and the 2−ΔΔCT method. Methods 2001, 25, 402–408. https://doi.org/10.1006/meth.2001.1262.

  1. Why authors did not employ other stress blood indicators, for example, lactate and glucose?

     Revising the bibliography available analyzing the effects of H2O2 exposure of different fish species allowed us to determine the plasmatic cortisol as the most reliable stress parameter in order to quantify fish stress response in this kind of exposure protocols. Other parameters, meanwhile, present variations and different response depending on the daytime exposure.

Roque, A., Yildiz, H. Y., Carazo, I., & Duncan, N. (2010). Physiological stress responses of sea bass (Dicentrarchus labrax) to hydrogen peroxide (H2O2) exposure. Aquaculture304(1-4), 104-107.

Vera, L. M., & Migaud, H. (2016). Hydrogen peroxide treatment in Atlantic salmon induces stress and detoxification response in a daily manner. Chronobiology International33(5), 530-542.

  1. Reference #34 refers to dairy cattle, not fish. Please replace this reference with a reference relevant to fish.

     We kindly thank this observation, a proper reference has been include.

  • De Verdal, H., Komen, H., Quillet, E., Chatain, B., Allal, F., Benzie, J. A., & Vandeputte, M. (2018). Improving feed efficiency in fish using selective breeding: a review. Reviews in Aquaculture10(4), 833-851.

  1. Line 136: The EU has suspended authorization for the use of ethoxyquin as a feed additive, so please explain why you used this additive.

     Thank you for pointing this mistake, it was an error on dietary formulation table foot note copy. The antioxidant employed in the dietary treatments was the BHT at a total concentration of 15 ppm.

  1. Line 122: Please mention the temperature range during the extrusion process for PHYTO0.1 and GMOS0.05 diets

We kindly apologize, but the enterprise in charge og diet manufacturing did not give us this information

  1. Line 150: Please clarify how 6-7 generations correspond to more than 35 years of selection efforts to facilitate the reader who is not familiar with the breeding selection

Thank you for pointing this possible misinformation, the materials and methods has been corrected in order to provide a more clear selective breeding protocol.

The full information is available in:

Montero, D., Carvalho, M., Terova, G., Fontanillas, R., Serradell, A., Ginés, R., Tuset, V., Acosta, F., Rimoldi, S., Bajek, A., Haffray, P., Allal, F. & Torrecillas, S. (2023). Nutritional innovations in superior European sea bass (Dicentrarchus labrax) genotypes: Implications on fish performance and feed utilization. Aquaculture572, 739486.

  1. Lines 173-175: Please refer to how many fish per tank were sampled.

     We want to thank the reviewer for pointing this misinformation, as its inclusion on the final manuscript will help clarifying the experimental procedures included.

Both, blood plasma and gill samples were taken as follow:

  • WT genotype: samples from 2 fish per tank, leading to a total of 6 samples per dietary treatment.
  • GS genotype: samples from 6 fish per tank, leading to a total of 6 samples per dietary treatment.

  1. English improvement:

  • Line 71: 'an intense oxidative agent'
  • Line 81: 'has been reported'
  • Lines 104-108: Improve English meaning
  • Line 181: 'and aeration'

     Thank you for identifying those mistakes on English language, the manuscript has been revised in order to improve the speech and grammar.

Reviewer 2 Report

Attached File

Author Response

  1. The aim of the study is unclearly defined, making it difficult to understand the purpose of the research. Additionally, there are some issues with the methodology, particularly in regard to the materials and methods used

We kindly thank the reviewer commentaries of the preliminary version of the present study, an important effort is going to be carried out in order to improve the expression of the different section of manuscript.

  1. Lines 112-124, the authors state that: “Four isonitrogenous and isoenergetic low FM/FO (10%/6%) based diets were formulated and produced by Biomar (Brande, Denmark), meeting all the described nutritional requirements for European sea bass juveniles”. However, it is unclear what criteria were used, or if there are any references to support these nutritional requirements.

     The reviewer was right about the misinformation related to the selection of the nutritional strategy employed, appropriate references have been included.

  • National Research Council. (2011). Nutrient requirements of fish and shrimp. National academies press.

  • Kousoulaki, K., Sæther, B. S., Albrektsen, S., & Noble, C. (2015). Review on European sea bass (Dicentrarchus labrax, Linnaeus, 1758) nutrition and feed management: a practical guide for optimizing feed formulation and farming protocols. Aquaculture Nutrition21(2), 129-151.

  1. Lin 126, it is Table 1 or Table S1? Please clarify / Line 232, Please confirm Table 2 or Table S2

     Thank you for pointing this detail, it is a mistake due to misunderstanding of Aquaculture Reports author’s guidelines. This error will be corrected, the tables in the manuscript will appear as “Table x” and the supplementary table (the one related to gene expression) will appear as “Table S1”

  1. There are inconsistencies in the manuscript regarding the scientific name of "Labiatae", and "Asteraceae" (Table S1) versus "Labiatae" and "Asteraceae" (Line 117, Abstract, etc.). Please ensure consistency throughout the manuscript. / Please ensure consistency in the manuscript for p value (p < 0.05) or (p<0.05).

     Thank you very much, these mistakes on text consistency have been correct.

  1. In the Experimental Conditions section, it is unclear what the authors are trying to express. Please provide a clearer explanation of this section. / In lines 207-210, it would be helpful to provide the quantity (mg) of gill sample for RNA extraction and the concentration (ng) of Total RNA for cDNA synthesis? / In line 232, it would be helpful to clarify whether the primers used in the study were designed by the authors or collected from previous publications, and if so, provide the references. / I recommend using "letter" to denote statistically significant differences among treatments in Table S4. In line 230, it is unclear whether t=0h refers to pre-treatment or post-treatment. Please clarify this point. / I recommend using "letter" to denote statistically significant differences among treatments in Table S4. / When evaluating the effectiveness of functional feed additives in promoting growth performance in fish, measuring the feed conversion ratio (FCR) is an important parameter to consider (Table 3).

     The material and methods section has been revised and better structured in order to clarify all the procedures and experimental conditions.

  1. It is extremely general in many parts, with the authors pointing out some areas of interest but not going into detail. The authors need to provide more detailed explanations of the relevance of their results, including how they support the aims of the study. Additionally, the comparison of current results with unrelated information is inappropriate, and the authors need to interpret the results more thoroughly, linking together the various parameters observed in the study.

     The discussion of results has been revised in order to make a better portray of the results obtained and the ideas expressed in the manuscript. Nevertheless, and despite we agree with the reviewer on the generality of the explanations provided, we would like to point that going more inside in the ideas proposed would be highly speculative as it is a still an underdeveloped field with, to our knowledge, a scarce variety of studies analyzing the combined effects of genetic selection and the supplementation of functional additives on low FM/FO based diets and their possible effects on fish stress tolerance and antioxidant status. Under our point of view, this study is a preliminary incursion in this field, indicating the possible biological mechanisms  and the direction of analyses in future studies.

Reviewer 3 Report

The article presents significant analysis and inferences regarding the use of additives in European sea bass, considering two different genotypes in conjunction with the application of hydrogen peroxide. This is a relevant and timely topic with many knowledge gaps. The introduction provides a well-presented overview. However, there are suggestions for improving and clarifying the contextual aspects related to the study within the manuscript.

Overall, the methodology is well-described and presented. However, this is also where my main criticism of the study lies. The statistical procedures employed are incorrect, leading to potential misinterpretation. For instance, conducting unidirectional analysis of variance does not allow for a comparison between genotypes. Merely using the initial weight as a covariate cannot account for genotypic differences in the observed responses, as weight itself is a phenotypic characteristic. Therefore, a two-way analysis of variance should be employed.

Furthermore, I am skeptical about using a three-way analysis of variance as the best approach for interpreting the results over time in relation to the two main parameters. Including time as a primary parameter can potentially lead to erroneous interpretations of the observed outcomes based on genotype and diet. My initial suggestion is for the authors to consider employing analysis methods related to repeated measures over time or exploring alternative multiple models to analyze the results. If the analysis is to be maintained as a three-way analysis, the results must be accurately presented as suggested in the manuscript. The reduction in the level of significance during the homoscedasticity analysis is a serious concern, and further discussion on this matter is warranted within the manuscript. I strongly recommend the authors consult a statistician expert to improve the analyses.

Given the criticism of the statistical analysis, it is apparent that the results and discussion sections can be significantly influenced. Despite this, suggestions have been provided to enhance the overall clarity and structure of the text in these sections.

Author Response

  1. Introduction improvements.

     We kindly thank the reviewer commentaries on this preliminary version of our manuscript, the introduction has been revised in order to better contextualize the background and the objectives of the present study. Additionally, an intensive revision of the English grammar and speech has been made in order to ensure the manuscript maximum quality.

  1. Suggestion: Meeting the nutritional requirements specified for European sea bass juveniles. Please, insert the reference about nutritional requirements used.

     Thank you for pointing this lack of information, two adequate references have been include.

  • National Research Council. (2011). Nutrient requirements of fish and shrimp. National academies press.

  • Kousoulaki, K., Sæther, B. S., Albrektsen, S., & Noble, C. (2015). Review on European sea bass (Dicentrarchus labrax, Linnaeus, 1758) nutrition and feed management: a practical guide for optimizing feed formulation and farming protocols. Aquaculture Nutrition21(2), 129-151.

  1. Could you kindly provide the detailed composition of the vitamin and mineral supplement as a footnote?

     We understand the vitamin and mineral composition of the dietary treatments is an important factor on experimental replicability, nevertheless giving this information compromises property rights from the enterprises involved on these dietary treatments production.

  1. Could you please elaborate on the rationale behind using a one-way ANOVA analysis instead of a two-way ANOVA analysis considering genotype and diet as the main parameters in this study? Despite the initial weight to be used as a co-variable, the results in Table S3 suggest apparently that the different genotypes show different responses. My suggestion is authors perform a two-way ANOVA to data of table s3.

     As the reviewer states, the final results obtained regarding fish growth performance were analyzed employing One-way ANOVA analysis and it is due to the experimental design of the study. The experience was designed in accordance to fish availability and initial fish body weight, leading to:

  • WT genotype – 360 fish distributed in 12 tanks (n= 3 tanks/ dietary treatment; 45 fish per tank)
  • GS genotype – 180 fish distributed in 4 tanks (n= 1 tank/ dietary treatment; 30 fish per tank)

     In the beginning of the feeding trial, inside each genotype level (GS or WT) no significant differences were found on fish initial body weight between the different experimental tanks.

  • WT IBW – F= 1.045 p-val= 0.405
  • GS IBW – F= 1.121 p-val= 0.344

Thus, the possible effects of the different dietary treatments on fish growth and feed utilization could be isolated in posterior analysis

     At the end of the feeding experience, inside each genotype, no significant effects of dietary treatment were found on fish final body weight (Image 2).

  • WT FBW – F= 2.306 p-val= 0.0758
  • GS FBW – F= 1.133 p-val= 0.339

     Regarding the possibility of performing Two-way ANCOVA or a TWO-way ANOVA analysis in order to evaluate fish performance, the statistical category GS only presented one tank per dietary treatment (n=1) making impossible the utilization of these statistical analysis. In addition, fish were not individually tagged at the beginning of the feeding trial (as it would suppose an elevated cost to analyse a secondary objective in the present study) making impossible to track fish individual performance.

     Thus, fish growth performance was analyzed by SGR individually for each statistical category (genotype or dietary treatment). The statistical analysis did not detected significant differences on fish SGR neither associated to fish genotype or the dietary treatment fed.

 SGR

Df

Sum Sq

Mean Sq

F value

Pr(>F)

genotype

1

0.0063

0.006302

1.662

0.218

Residuals

14

0.05309

0.003792

 SGR

Df

Sum Sq

Mean Sq

F value

Pr(>F)

diet

3

0.00632

0.002106

0.192

0.9

Residuals

12

0.13142

0.010952

Additionally, and in order to provide a better discussion on fish growth performance, the same analysis are going to be included for fish FCR and feed intake (total FI and Individual FI calculated as: total feed consumption/ tank final fish number).

 FCR

Df

Sum Sq

Mean Sq

F value

Pr(>F)

genotype

1

0.2626

0.2626

8.335

0.0119

Residuals

14

0.441

0.0315

GS gnotpe

WT  genotype

FCR mean

=

1.53

1.82

FCR sd

=

0.06

0.2

 FCR

Df

Sum Sq

Mean Sq

F value

Pr(>F)

diet

3

0.0724

0.02412

0.459

0.716

Residuals

12

0.6312

0.0526

 Total FI

Df

Sum Sq

Mean Sq

F value

Pr(>F)

genotype

1

315006

315006

17.9

0.000839

Residuals

14

246369

17598

GS genotype

WT genotype

Total FI mean

=

3736.235

3412.195

Total FI sd

=

103.1824

139.6194

 Total FI

Df

Sum Sq

Mean Sq

F value

Pr(>F)

diet

13427

4476

0.098

0.96

Residuals

547948

45662

 Indiv FI

Df

Sum Sq

Mean Sq

F value

Pr(>F)

genotype

1

7499

7499

1105

1.01E-14

Residuals

14

95

7

GS genotype

WT genotype

Indiv. FI mean

=

126.68

76.68

Indiv. FI sd

=

 1.58

2.82

 Indiv FI

Df

Sum Sq

Mean Sq

F value

Pr(>F)

diet

3

8

2.7

0.004

0.99

Residuals

12

7586

632.2

  1. The statement regarding the statistical procedure raises some concerns. While it is commendable that the data were tested for normality and homogeneity, the approach of reducing the alpha level to 0.01 when homogeneity assumptions are violated is questionable.

     We thank the reviewers high quality comments made on the statistical analysis reliability. We would like to point the robustness of the analysis performed and the different decisions made in the statistical analysis.

  • All data analyzed were tested for outlying values through linear regression adjustment, defining the outside cut-offs as 1.5 times the Inter-Quantile Range (IQR) below the first and above the third quantiles [Hoaglin and Iglewicz, 1987; Feng and Zou, 2008; Blanca et al., 2017; Blanca et al., 2018].

  • The F-test α-value at 0.05 and before performing any analysis both normality of data and variance homogeneity was tested. When the data did not accomplished one of both ANOVA requirements, the data were transformed in order to pursue criteria accomplishment. First, the sqrt (x +1) and in case it did not worked the log(x+1) transformation. In case none of the transformations were able to accomplish ANOVA criteria, the analysis were repeated with the data without transformation and setting the F-test α-value at 0.01 because ANOVA is robust to departures from this assumption, in which the total sample n> 30 and balanced [Forcada et al., 2009; Forcada et al., 2010; Blanca et al., 2017; Blanca et al., 2018].

      Circulating plasma cortisol: n= 6 x 2 genotypes x 4 diets x 3 times -> ntotal = 72

      Gill relative gene express:    n= 6 x 2 genotypes x 4 diets x 3 times -> ntotal = 72

      WT fish initial weight:           n= 45 fish x 4 diets x 3 tanks -> ntotal = 540

      GS fish initial weight:            n= 30 fish x 4 diets x 1 tank -> ntotal = 120

      WT fish final weight:             n≈ 45 fish x 4 diets x 3 tanks -> ntotal ≈ 540

      GS fish initial weight:            n≈ 30 fish x 4 diets x 1 tank -> ntotal ≈ 120

  • Those analysis with lower total samples n, such as SGR,FCR and FI were also tested for normality and homogeneity and none assumptions were broken. Thus, non-parametric analysis such as Kruskall-Wallis or PERMANOVA were not required.

  • Blanca Mena, M. J., Alarcón Postigo, R., Arnau Gras, J., Bono Cabré, R., & Bendayan, R. (2017). Non-normal data: Is ANOVA still a valid option?. Psicothema, 2017, vol. 29, num. 4, p. 552-557.

  • Blanca, M. J., Alarcón, R., Arnau, J., Bono, R., & Bendayan, R. (2018). Effect of variance ratio on ANOVA robustness: Might 1.5 be the limit?. Behavior Research Methods50, 937-962.

  • Forcada, A., Valle, C., Sánchez-Lizaso, J. L., Bayle-Sempere, J. T., & Corsi, F. (2010). Structure and spatio-temporal dynamics of artisanal fisheries around a Mediterranean marine protected area. ICES Journal of Marine Science67(2), 191-203.

  • Forcada, A., Valle, C., Bonhomme, P., Criquet, G., Cadiou, G., Lenfant, P., & Sánchez-Lizaso, J. L. (2009). Effects of habitat on spillover from marine protected areas to artisanal fisheries. Marine Ecology Progress Series379, 197-211.

  • Feng, M., Ll, Q., & Zou, Z. (2008). An outlier identification and judgment method for an improved neural‐network BOF forecasting model. steel research international, 79(5), 323-332.

  • Hoaglin, D. C., & Iglewicz, B. (1987). Fine-tuning some resistant rules for outlier labeling. Journal of the American statistical Association82(400), 1147-1149.

  1. The presentation and organization of the results in the table for the three-way ANOVA are not appropriate. To provide a reference for a correctly formatted table, I recommend reviewing an example from Chung et al. (2020).

     Thank you for providing this example of three way analysis results report, it has been truly helping in order to present our results.

  1. The text lacks a clear and concise structure. It would be helpful to divide the discussion into subsections to address different aspects of the findings, such as the impact of selective breeding, the role of cortisol responsiveness, and the effects of dietary supplementation on antioxidant status.

     The discussion has been revised and more concisely written, adding a better structure of results presentation. We kindly thank the suggestion of speech precision in order to increase the manuscript clarity and quality.

Round 2

Reviewer 2 Report

- Please check again for the format from Lines 82-87,

- Table 1: why were the data changed compared to the previous version? 

- Lines 255-258: target genes were mentioned above. I suggest removing it.

 -Please indicate the letters used in Table 3. I recommend using small letters. Same as for Table 4.

- Please re-check for H2O2 and O2 in the manuscript, especially in the discussion section.

- Please re-check the scientific gene names used in this study, especially in the discussion section

Author Response

We want to thank the reviewer for the accurate commentaries and suggestions on manuscript review process. We have included the last correction to improve our study.

 Please check again for the format from Lines 82-87

The lines have been rewritten in order to better express the ideas exposed in the introduction.

- Table 1: why were the data changed compared to the previous version? 

The previous dietary composition was incorrect, as we described an old formula employing ethoxyquin as antioxidant, which is banned by the EU. It was a copy-paste error. On the first review process we actualized the data presenting the formula used in the experience, in which the antioxidant employed was the BHT at a final concentration of 150 ppm.

- Lines 255-258: target genes were mentioned above. I suggest removing it.

Table 2 foot note has been actualized and the target gene list removed as indicated by the reviewer.

 -Please indicate the letters used in Table 3. I recommend using small letters. Same as for Table 4.

Differences on growth parameters, circulating plasma cortisol levels and gill gene expression levels(Supplementary table) have been expressed in small letters, as recommended by the reviewer.

- Please re-check for H2O2 and O2 in the manuscript, especially in the discussion section.

We apologize by the lack of consistency on H2O2 and O2 nomenclature, it has been corrected.

- Please re-check the scientific gene names used in this study, especially in the discussion section

On the same way the scientific name of the employed genes has been revised and corrected.

Reviewer 3 Report

There are no more suggestions to the authors.

Author Response

We want to thank the reviewer for the effort made reviewing our manuscript in order to increase the final quality of our study